# Neural dynamics of perceptual inference and its reversal during imagery

**Nadine Dijkstra[1,2]\*, Luca Ambrogioni[1], Diego Vidaurre[3,4], Marcel van Gerven[1]**

[1]Donders Centre for Cognition, Radboud University, Donders Institute for Brain, Cognition and Behaviour, Nijmegen, Netherlands; [2]Wellcome Centre for Human Neuroimaging, University College London, London, United Kingdom; [3]Oxford Centre for Human Brain Activity, Oxford University, Oxford, United Kingdom; [4]Department of Clinical Health, Aarhus University, Aarhus, Denmark

**Abstract** After the presentation of a visual stimulus, neural processing cascades from low-level sensory areas to increasingly abstract representations in higher-level areas. It is often hypothesised that a reversal in neural processing underlies the generation of mental images as abstract representations are used to construct sensory representations in the absence of sensory input. According to predictive processing theories, such reversed processing also plays a central role in later stages of perception. Direct experimental evidence of reversals in neural information flow has been missing. Here, we used a combination of machine learning and magnetoencephalography to characterise neural dynamics in humans. We provide direct evidence for a reversal of the perceptual feed-forward cascade during imagery and show that, during perception, such reversals alternate with feed-forward processing in an 11 Hz oscillatory pattern. Together, these results show how common feedback processes support both veridical perception and mental imagery.

## Introduction

When light hits the retina, a complex cascade of neural processing is triggered. Light waves are transformed into electrical signals that travel via the lateral geniculate nucleus of the thalamus to the visual cortex (*Card and Moore, 1989*; *Reid and Alonso, 1995*) (but see [*Bullier, 2001*] for other routes). First, low-level visual features such as orientation and spatial frequency are processed in primary, posterior visual areas (*Hubel and Wiesel, 1968*) after which activation spreads forward towards secondary, more anterior visual areas where high-level features such as shape and eventually semantic category are processed (*Maunsell and Newsome, 1987*; *Thorpe and Fabre-Thorpe, 2001*; *Vogels and Orban, 1996*). This initial feed-forward flow through the visual hierarchy is completed within 150 ms (*Seeliger et al., 2018*; *Thorpe et al., 1996*) after which feedback processes are assumed to further sharpen representations over time until a stable percept is achieved (*Cauchoix et al., 2014*; *Kok et al., 2012*).

Activation in visual areas can also be triggered internally, in the absence of external sensory signals. During mental imagery, information from memory is used to generate rich visual representations. Neural representations activated during imagery are highly similar to those activated during perception (*Dijkstra et al., 2019*). Imagining an object activates similar object representations in high-level visual cortex (*Dijkstra et al., 2017*; *Ishai et al., 2000*; *Lee et al., 2012*; *Reddy et al., 2010*) and generating a mental image with simple visual features such as oriented gratings or letters is associated with perception-like activation of low-level visual areas (*Albers et al., 2013*; *Pearson et al., 2008*; *Senden et al., 2019*).

In contrast, the temporal dynamics underlying the activation within the visual system during perception and mental imagery are presumably very different. The neural dynamics of the early stages of perception have been extensively characterised with intracranial electrophysiological recordings

**\*For correspondence:**
n.dijkstra@donders.ru.nl

**Competing interests:** The authors declare that no competing interests exist.

in primates (*Thorpe et al., 1996*; *Thorpe and Fabre-Thorpe, 2001*; *Hubel and Wiesel, 1968*). How-ever, the neural dynamics of imagery, i.e. how activation travels through the brain during internally generated visual experience, remain unclear. Researchers from various fields have proposed that the direction of information flow during internally generated visual experience is reversed compared to perception (*Ahissar and Hochstein, 2004*; *Hochstein and Ahissar, 2002*; *Kosslyn et al., 2001*; *Pearson and Keogh, 2019*). In line with this idea, a recent study showed that during memory recall, high-level, semantic representations were active before low-level, perceptual representations (*Linde-Domingo et al., 2019*). However, the localisation of activation in this study was ambiguous such that it is possible that all processing happened within high-level visual cortex but that only the dimension to which the neurons were sensitive changed from abstract features to perceptual features over time (for an example of dynamic neural tuning, see *Spaak et al., 2017*. Moreover, memory recall was not directly compared with memory encoding. Therefore, it remains unclear whether the same percep-tual cascade of neural activation is reactivated in reverse order during internally generated visual experience.

According to predictive processing (PP) theories, reversals of information flow also play an impor-tant role during perception. PP states that the brain deals with the inherent ambiguity of incoming sensory signals by incorporating prior knowledge about the world (*Helmholtz, 1925*). This knowl-edge is used to generate top-down sensory predictions which are compared to the bottom-up sen-sory input. Perceptual inference is then accomplished by iteratively updating the model of the world until the difference between prediction and input is minimised (*Friston, 2005*; *Knill and Pouget, 2004*). Therefore, the neural dynamics of stimulus information during perception should be charac-terised by an interplay between feed-forward and feedback sweeps. Simulations based on PP mod-els predict that these recurrent dynamics are dominated by slow-wave oscillations (*Bastos et al., 2012*; *Lozano-Soldevilla and VanRullen, 2019*).

In this study, we used magnetoencephalography (MEG) and machine learning to characterise the spatio-temporal dynamics of information flow during mental imagery and perception. We first char-acterised neural activity during the initial perceptual feed-forward sweep using multivariate classifiers at different time points which served as proxies for representations in different visual areas. That is, decoding at early perception time points was taken to reflect stimulus representations in low-level, posterior visual areas while decoding at later time points was taken to reflect high-level, anterior visual representations. Then, we estimated when these feed-forward perception models were reacti-vated during imagery and later stages of perception to infer the neural dynamics of information processing.

## Results

Twenty-five participants executed a retro-cue task while MEG was measured. During the task, two consecutive stimuli were presented, a face and a house or a house and a face, followed by a cue indicating whether participants had to imagine the first or the second stimulus. They then imagined the cued stimulus as vividly as possible and indicated their experienced imagery vividness. There were eight exemplars per stimulus category which were chosen to be highly similar to minimise vari-ation in low-level details within categories and to maximise between-category differences. To ensure that participants were generating detailed mental images, we included catch trials during which par-ticipants had to indicate which of four highly similar exemplars they had just imagined. An accuracy of 89.9% (*SD* = 5.4%) indicated that participants did indeed imagine the stimuli with a high degree of visual detail.

### Inferring information flow using perceptual feed-forward classifier models

Classifier models representing neural representations at different time points during perception were obtained using linear discriminant analysis (LDA). An LDA classifier was trained to decode the stimulus class ('face' vs 'house') from sensor level activity at each time point, giving different percep-tion models for different time points (*Figure 1A*). We decided to only focus on the time period between 70 ms and 130 ms after stimulus onset, which, with a sampling rate of 300 Hz, contained 19 perception models. Perceptual content could be decoded significantly better than chance level at 70 ms after stimulus onset (*Dijkstra et al., 2018*). This is in line with intracranial and MEG studies

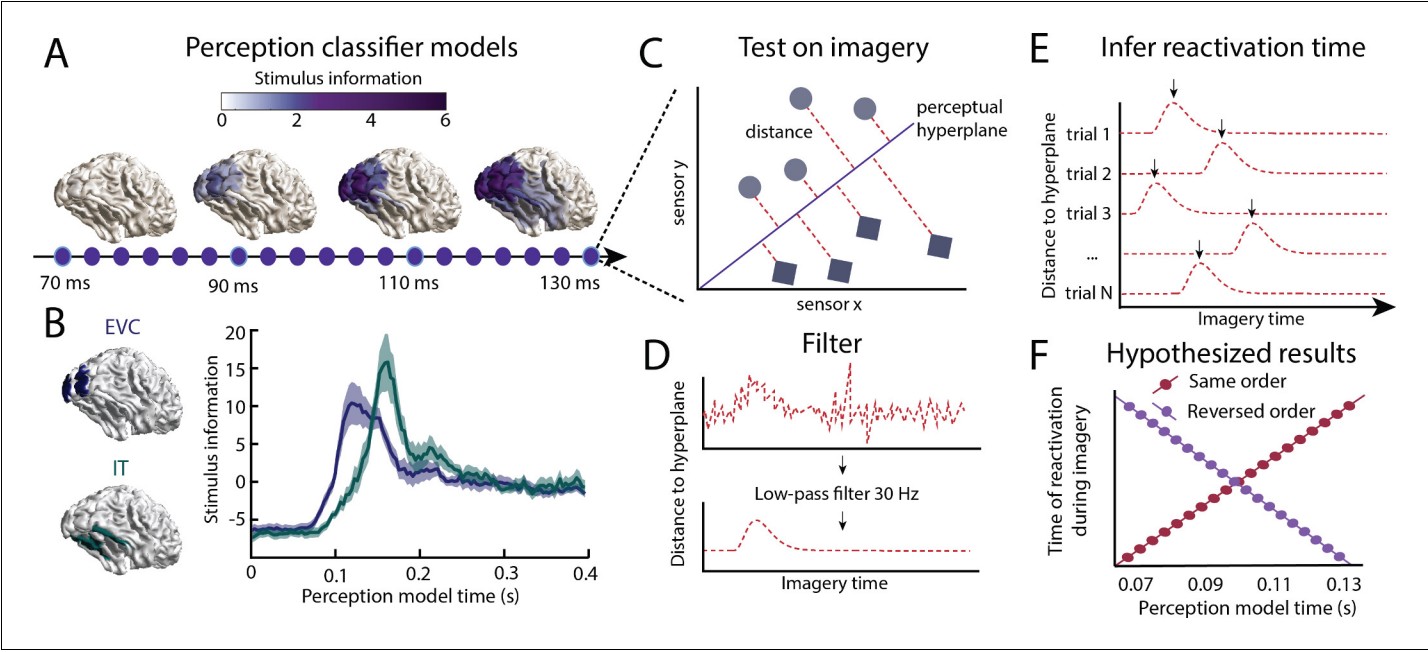

**Figure 1.** Inferring information flow using perceptual feed-forward classifier models. (A–B) Perception models. At each point in time between 70 and 130 ms after stimulus onset, a perception model (classifier) was estimated using Linear Discriminant Analysis (LDA) on the activation patterns over sensors. (A) The source-reconstructed difference in activation between faces and houses (i.e. decoding weights or stimulus information) is shown for different time points during perception. (B) Stimulus information standardised over time is shown for low-level early visual cortex (EVC: blue) and high-level inferior temporal cortex (IT: green). These data confirm a feed-forward flow during the initial stages of perception. (C) Imagery reactivation. For each trial and time point during imagery, the distance to the perceptual hyperplane of each perception model is calculated. (D) To remove high-frequency noise, a low-pass filter is applied to the distance measured. (E) The timing of the reactivation of each perception model during imagery is determined by finding the peak distance for each trial. (F) Hypothesised results. This procedure results in a measure of the imagery reactivation time for each trial, for each perception model time point. If perception models are reactivated in the same order during imagery, there would be a positive relation between reactivation imagery time and perception model time. If instead, perception models are reactivated in reverse order, there would be a negative relation. Source data associated with this figure can be found in the *Figure 1—source data 1* and *2*.

The online version of this article includes the following source data for figure 1:

**Source data 1.** Source data for panel A.
**Source data 2.** Source data for panel B.

which showed that visual information is detectable in early visual cortex from 50 ms onwards (*Thorpe and Fabre-Thorpe, 2001*; *Ramkumar et al., 2013*). Furthermore, both intracranial as well as scalp electrophysiology has shown that around 130 ms, high-level object representations first get activated (*Isik et al., 2014*; *Maunsell and Newsome, 1987*; *Seeliger et al., 2018*; *Vogels and Orban, 1996*). Therefore, this early time window is representative of the feed-forward sweep during perception. In line with this, source reconstruction of the sensor-level activation patterns at different time points shows that stimulus information (i.e. the difference in activation between faces and houses [*Haufe et al., 2014*] spreads from low-level visual areas towards higher-level visual areas during this period (*Figure 1A*). Furthermore, cross-correlation between the information flow in early visual cortex (EVC) and inferior temporal cortex (IT; *Figure 1B*) confirms that stimulus information is available in low-level EVC 26.3 ms (*CI* = 13.1 to 32.9 ms) before it reaches high-level IT.

To identify when these representations were reactivated during imagery, we tested the perception models on imagery to obtain the distances to the classifier hyperplanes per trial (*Figure 1C–E*). The distance to the hyperplane indicates the amount of classifier evidence present in the data. Distance measures have previously been used as a measure of model activation (*Kerrén et al., 2018*; *Linde-Domingo et al., 2019*) and have been linked to reaction time measurements (*Grootswagers et al., 2018*). For each perception model and for each imagery trial, we identified the time of the absolute peak distance (*Linde-Domingo et al., 2019*). This resulted in a trial-by-trial estimate of the reactivation timing for the different perception models. If processing happens in a

similar order during imagery as during perception, we would expect that during imagery, early perception models are reactivated earlier in time than late perception models (*Figure 1F*). This would result in a positive relation between perception model time and imagery reactivation time. If instead, processing happens in reverse order, with late, high-level models being active before earlier models, we would see a negative relation between perception model time and imagery model time.

We tested whether this analysis approach was indeed able to infer the order of reactivation of neural representations using simulations (see Methods; Simulations). The perceptual feedforward sweep was simulated as the sequential activation of five neural representations, operationalised as pseudorandom sensor projections (*Figure 2A*). We then tested whether distance measures could successfully identify the order of reactivation in a testing (i.e. imagery) set with either small temporal jitter (SD of the onset between trials = 0.1 s) or large temporal jitter (SD = 0.5 s). First the cross-decoding accuracy is plotted for each combination of training and testing time point, computed by averaging classifier performance over trials per time point (*Figure 2B–D*; top panels). For small amounts of temporal jitter, the temporal profile of the decoding accuracy clearly differentiates between same or reversed order of reactivation (compare top-left panel of *Figure 2B* and *Figure 2C*). However, for larger temporal jitter, which is supposedly the case during mental imagery (*Dijkstra et al., 2018*), the temporal profile of the decoding accuracy smears out over the testing time axis, completely obscuring the order of reactivations. In contrast, the reactivation time defined as the peak distance time point per trial accurately indicates the direction of the relationship with the training sequence, irrespective of the amount of temporal jitter.

Our approach has a close relationship with temporal generalisation decoding analysis (*King and Dehaene, 2014*). Temporal generalisation characterises the stability of neural representations over time by training and testing a classifier on different time points of either the same or different conditions, resulting in a time by time decoding accuracy matrix. Above chance accuracy between two different time points indicates that the neural representation of the stimulus at those time points is similar, whereas chance decoding indicates that the representation between two time points is not similar and has therefore changed over time. In our approach we also train and test classifiers at different time points. However, instead of computing the accuracy between different training and testing time-points by averaging over trials per time point, we determine per training time point and

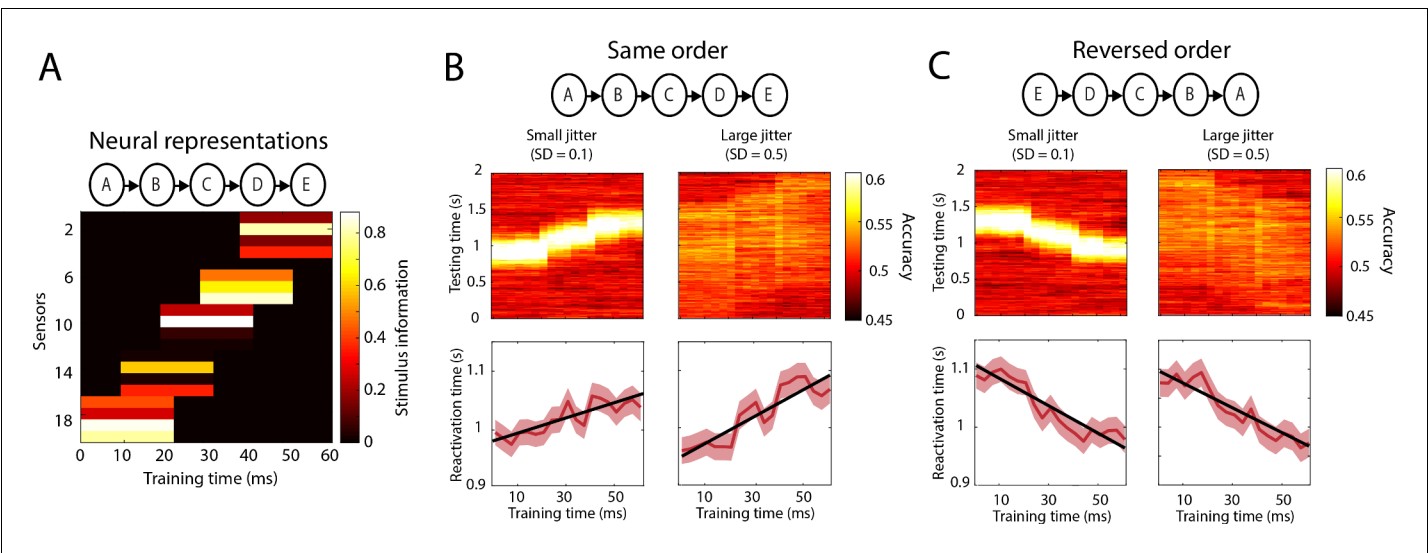

**Figure 2.** Simulations reactivation analysis. (A) Five different neural representations, activated sequentially over time, were modelled as separate sensor activation patterns. Stimulus information is defined as the difference in activation between the two classes (i.e. what is picked up by the classifier). Results on a testing data-set in which the order of activation was either the same (B) or reversed (C). Top panels: cross-decoding accuracy obtained by averaging over trials. Bottom panels: reactivation times inferred per trial. Source data associated with this figure can be found in the *Figure 2—source data 1*.

The online version of this article includes the following source data for figure 2:

**Source data 1.** Source data from the simulations.

per trial, which testing time point is most similar to the training time point, indicated by the time point with the maximum LDA distance in favour of the correct class (*Linde-Domingo et al., 2019*; *Kerrén et al., 2018*; *Grootswagers et al., 2018*). In other words, our method was targeted at the ordering of the representations, and, critically, did not assume a consistent timing between trials, which has been shown to be a limitation of the temporal generalisation decoding scheme (*Vidaurre et al., 2019*). When applied to different time epochs, our method revealed whether the order of activation of neural representations is the same or reversed while at the same time allowing for variation in the exact onset of reactivation between trials. We re-emphasise that this is especially important in the current context given that the timing of imagery likely substantially differs between trials, which means that the temporal dynamics will be obscured when computing accuracy by averaging over trials per time point (*Dijkstra et al., 2018*).

## Perceptual feed-forward sweep is reversed during imagery

The reactivation time during imagery for the different perception models is shown in *Figure 3A*. To test whether there was a significant ordering in the reactivation, we ran a linear mixed-effects model (LMM) with the reactivation time during imagery as dependent variable, the perception model time as fixed predictor, and subject and trial as random variables. Five models with different combinations of random effects were estimated and the model with the highest Schwarz Bayesian Information Criterion (BIC) was used to ensure best-fit with minimum number of predictors (*Supplementary file 1a*). The winning model contained a random effect for the intercept of each subject and trial. This means that this model allowed the reactivation of the perception sequence to start at different time points per trial and per subject, which is in line with the idea that there is a large variation in timing of imagery between trials and subjects (*Dijkstra et al., 2018*).

The model showed a significant main effect of perception time ($t(98160) = -5.403$, p=6.58e-8) with a negative slope ($\beta0 = 2.05$, $SD = 0.28$; $\beta1 = -1.17, SD = 0.22$) indicating that models of later perception time points were associated with earlier imagery reactivation times. The fact that the absolute slope is so close to one suggests that reactivation during imagery of the perception model sequence happens at a similar speed as the original activation during perception.

Next, we reconstructed the imagery activation by realigning the trials based on the identified peak time points for each perception model time point. The imagery time line was inferred using the linear equation obtained from the LMM. The temporal dynamics of high-level IT and low-level EVC (*Figure 3B*) confirm the conclusion that during imagery, information flows from high-level to low-level visual areas. Cross-correlation between these realigned signals shows that information in IT precedes information in EVC by 11.2 ms ($CI = 0$ to 29.9 ms). Furthermore, before realignment, time-locked decoding during imagery only revealed a small amount of information in high-level visual cortex (*Figure 3D*). In contrast, after realignment, stimulus information was also clearly present in EVC. This emphasises how time-locked analyses obscure neural processing during complex cognitive processes such as mental imagery.

To ensure that these results were not due to confounds in the structure of the data irrelevant to reactivation of stimulus representations, we performed the same analysis after permuting the class-labels of the trials, erasing stimulus information while keeping the temporal structure of the data unaltered. Specifically, we trained the perception models using random class assignments and then again calculated the reactivations during imagery. The results are shown in *Figure 3B*. The sequential reactivation disappeared when using shuffled classifiers as perception time did not significantly predict imagery reactivation time anymore ($t(98160) = -0.762$, p=0.446). Furthermore, for the main analysis, we removed high frequency noise from the imagery distance traces. To check whether this filter somehow altered the results, we ran the same analysis without the low-pass filter, which gave highly similar results (see *Figure 3—figure supplement 1*). Without the filter, there was still a significant main effect of perception time ($t(98160) = -4.153$, p=0.0000328) where models of later perception time points were associated with earlier imagery reactivation times ($\beta=-0.8942$, SD = 0.2153) and the results of the shuffled classifier remained non-significant. Finally, because we used cross-validation within subjects to calculate reactivation timing, there is a dependence between trials of the same subject, violating the independence assumption of first-order statistical tests. As a sanity check, we performed a second test that did not require splitting trials: we used LMM models with reactivation times averaged over trials within subjects, and only 'subject' as a random variable. In this case, the model containing both a random effect of intercept as well as slope per subject best

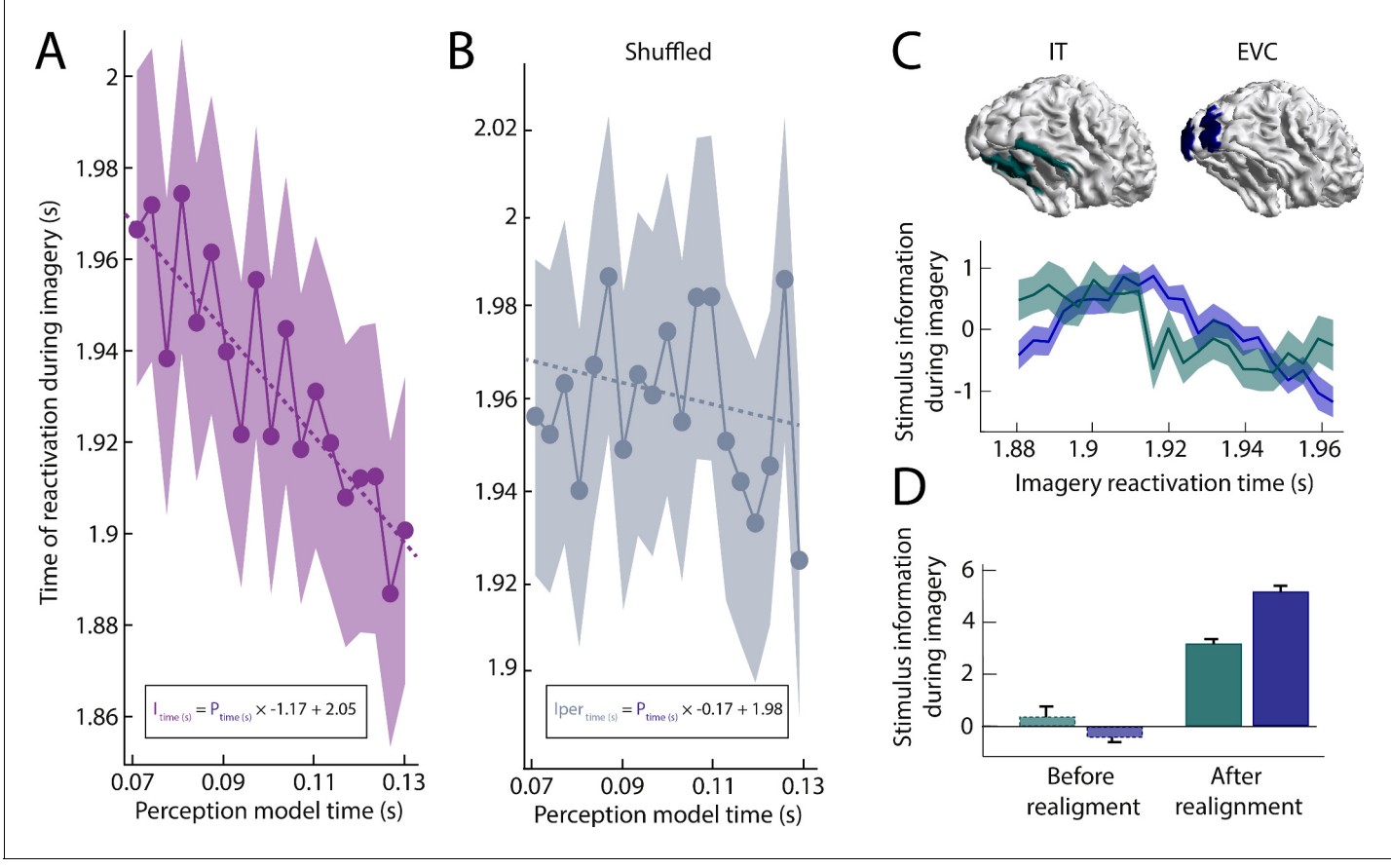

**Figure 3.** Imagery reactivation results. (**A**) Reactivation time during imagery for every perception model averaged over trials. The shaded area represents the 95% confidence interval. The linear equation shows how imagery reactivation time (I) can be calculated using perception model time (P) in seconds. (**B**) Same results after removing stimulus information by permuting the class-labels (**C–D**) Stimulus information during imagery was estimated by realigning the trials based on the reactivation time points and using the linear equation to estimate the imagery time axis. (**C**) Stimulus information standardised over time for low-level early visual cortex (EVC) and high-level inferior temporal cortex (IT). (**D**) Stimulus information in EVC and IT averaged over time before and after realignment. Stimulus information below 0 indicates that the amount of information did not exceed the permutation distribution. Source data associated with this figure can be found in the *Figure 2—source data 1–6*.

The online version of this article includes the following source data and figure supplement(s) for figure 3:

**Source data 1.** Contains the reactivation time per perception model and per trial, the subject ID ('S').

**Source data 2.** Contains the source-reconstructed stimulus information as subject x source parcel x time points as well as the corresponding time vector and source-parcel names for the realigned data.

**Source data 3.** Contains the source-reconstructed stimulus information as subject x source parcel x time points as well as the corresponding time vector and source-parcel names for the unaligned data.

**Source data 4.** Contains the reactivation time per perception model and per trial, the subject ID ('S') for the permuted classifier.

**Source data 5.** Contains the reactivation time per perception model and per trial, the subject ID ('S') for the unfiltered data.

**Source data 6.** Contains the reactivation time per perception model and per trial, the subject ID ('S') for the unfiltered and permuted data.

**Figure supplement 1.** Imagery reactivation results without the low-pass filter.

explained the data (see *Supplementary file 1c*). This between-subject model still showed a significant main effect of perception time (*t*(24) = −3.24, p=0.003) with a negative slope (β0 = 2.05, *SD* = 0.03, β1 = −1.19, *SD* = 0.37) confirming that the effect was not dependent on between-trial statistics.

## Reactivation during imagery reveals recurrent processing during later stages of perception

In the previous analysis, we focused on the first 150 ms after stimulus presentation because this period reflects the initial perceptual feed-forward sweep. A negative relationship with imagery

reactivation time therefore indicated feedback processing during imagery (*Figure 1*). However, feedback processes are assumed to play a fundamental role in later stages of perception (*Lamme and Roelfsema, 2000*; *Pennartz et al., 2019*). For these later stages of perception, we would therefore expect a positive relation with imagery reactivation, indicating that information flows in the same direction. To investigate this, we next calculated the imagery reactivation time for all time points during perception (*Figure 4*). The results between 70 ms and 130 ms are equivalent to *Figure 3A*.

Interestingly, during the first 400 ms of perception, there seems to be an oscillatory pattern in the relationship between perception time and imagery reactivation time, where positive and negative slopes alternate. This suggests that during perception, the direction of information flow alternates. This pattern repeats four times in 400 ms, roughly reflecting an alpha oscillation. To investigate this further, we quantified 10 Hz power over time using a Morlet decomposition (*Figure 4B* left, purple curve) and compared the results with the permuted classifier (*Figure 4B* left, grey curve). There was a significant increase in 10 Hz power between 80 and 315 ms after stimulus onset during perception (all FDR corrected *p*-values below 0.003). Furthermore, a Fast Fourier Transform over the first 400

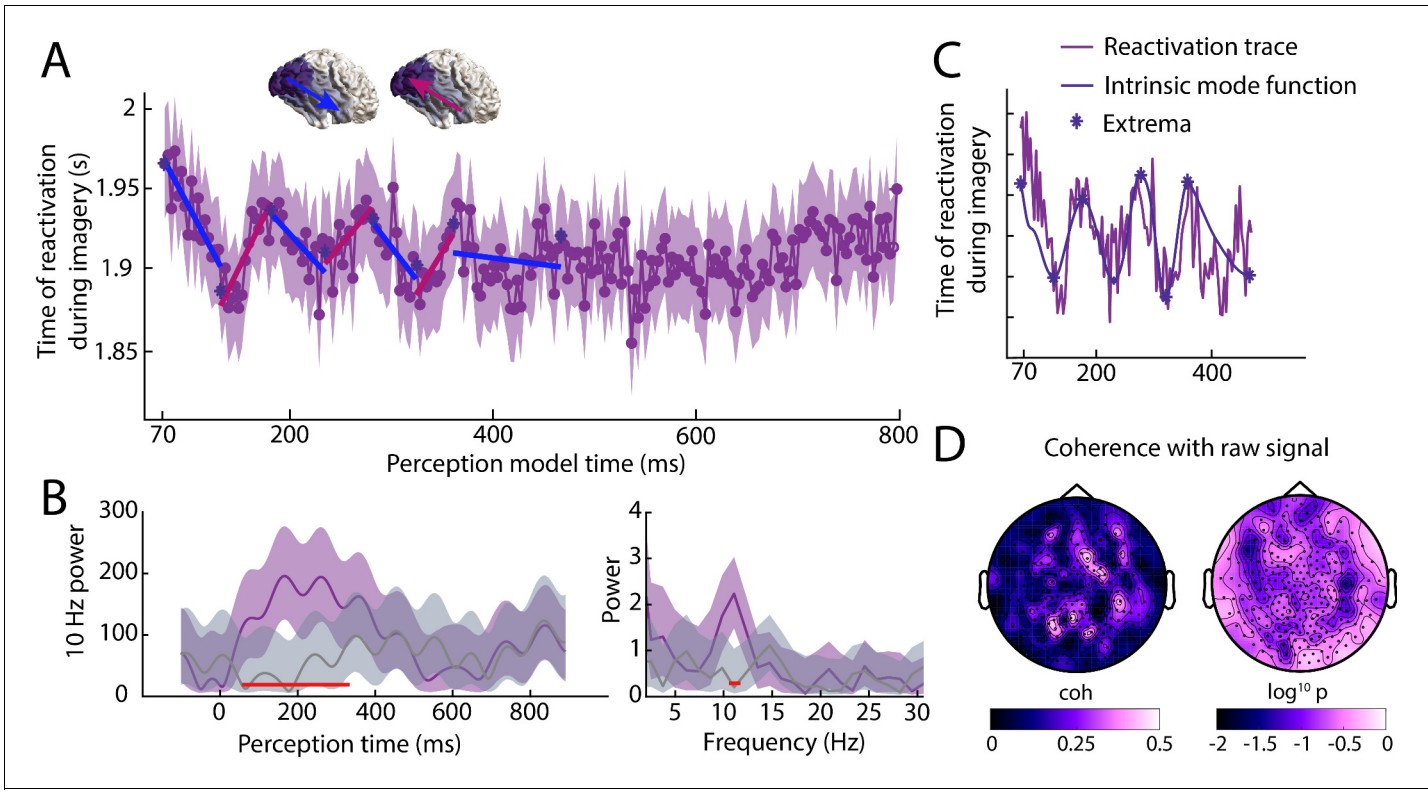

**Figure 4.** Reactivation timing during imagery for classifiers trained at all perception time points. (**A**) Imagery reactivation time for perception models trained on all time points. On the x-axis the training time point during perception is shown and on the y-axis the reactivation time during imagery is shown. The dots represent the mean over trials for individual time points and the shaded area represents the 95% confidence interval. (**B**) Left: time-frequency decomposition using a Morlet wavelet at 10 Hz. Right: power at different frequencies using a Fast Fourier Transformation. The purple line represents the true data and the grey line represents the results from the shuffled classifier. Shaded areas represent 95% confidence intervals over trials. Red lines indicate time points for which the true and shuffled curve differed significantly (FDR corrected). (**C**) Intrinsic mode function and its extrema derived from the reactivation traces using empirical model decomposition. (**D**) Coherence between reactivation trace and raw signal. Left coherence values, right log10 p values. Source data associated with this figure can be found in the *Figure 4—source data 1–3*.

The online version of this article includes the following source data and figure supplement(s) for figure 4:

**Source data 1.** Contains the identified reactivation trace extrema based on EMD of the empirical traces.
**Source data 2.** Contains the reactivation traces as reactivation sample point per trial x perception model time for the unpermuted classifiers.
**Source data 3.** Contains the reactivation traces as reactivation sample point per trial x perception model time for the permuted classifiers.
**Figure supplement 1.** Reactivation timing during imagery for all perception time points without low-pass filter.
**Figure supplement 2.** Dynamics of the raw signal during perception for occipital channels.
**Figure supplement 3.** Imagery reactivation traces for the full perception period separately for each subject.

ms revealed that the difference in power was limited to the 11.2 Hz frequency (*Figure 4B* right, p=0.001).

These results indicate that, during perception, stimulus information travels up and down the visual hierarchy aligned to the alpha frequency. To make sure this effect was not due to the low-pass filtering, we applied the same analysis on the unfiltered data, giving almost identical results (see *Figure 4—figure supplement 1*). We next aimed to parse the signal into feed-forward and feedback sweeps. To this end, we used empirical mode decomposition (EMD) over the first 400 ms to separate the signal in intrinsic mode functions (IMFs). We selected identified increasing and decreasing phases by selecting the period between two subsequent extrema (*Figure 4C*) and calculated the slope for each phase (single estimated slopes are shown in *Figure 4A*). The average decreasing (feed-forward) slope was −0.69 (*CI* = −2.05 to −0.35) and the average increasing (feedback) slope was 1.02 (*CI* = −0.31 to 2.14; *Figure 4D*). There was no significant difference in the absolute slope value between increasing and decreasing phases (*M*diff = 0.33, *CI*: −1.80 to 1.74), revealing no evidence for a difference in processing speed for feed-forward and feedback sweeps.

It might be possible that the oscillation in the reactivation trace is caused by evoked alpha oscillations in the raw signal which modulates signal-to-noise ratio via its amplitude. As can be seen in *Figure 4—figure supplement 2*, there is indeed a clear evoked alpha oscillation present in the raw signal. To investigate whether the oscillation of the reactivation trace was related to the oscillation in the raw signal, we calculated the spectral coherence at the peak frequency (*Figure 4B*) between the reactivation trace and the raw signal at each sensor. We then compared the coherence between the raw data and the true activation reactivation with the coherence between the raw data and the reactivation of the shuffled classifier using a permutation test with 10000 permutations. None of the sensors showed a significant difference in coherence between the true and permuted classifier (FDR corrected; *Figure 4D*), even though the reactivation was only present for the true classifier (*Figure 4B*). Furthermore, the topography of the coherence values appears very unstructured (*Figure 4D*). Together, this suggests that the oscillatory pattern in the reactivated trace does not merely reflect evoked alpha in the raw signal. However, future research is necessary to fully characterise the relationship between the reactivation dynamics and the raw signal dynamics.

## Stimulus representations are iteratively updated during perception

If there is indeed a recurrent information flow up and down the visual hierarchy during perception, we should also be able to demonstrate this within perception. The previous results suggest specific time windows of feed-forward and feedback phases during perception. To test whether these different phases indeed reflected reversals of information flow, we applied the reactivation analysis previously used for imagery (*Figure 1*) to the different perception phases identified in the previous analysis (*Figure 4*). We predicted that perception classifiers trained during a decreasing (feed-forward) phase would be reactivated in reverse order during an increasing (feedback) phase and vice versa, showing a negative relationship between training time and reactivation time. In contrast, classifiers trained and tested on the same type of phase (both decreasing or both increasing), should show a positive relationship (see *Figure 5B*, for hypothesised results).

The reactivation traces for the different test phases are shown in *Figure 5A*. Blue traces represent reactivations during feed-forward phases and pink traces represent reactivations during feedback phases. Grey traces show the results for a permuted classifier. In line with the previous findings, for most phases, there is a clear oscillatory pattern between training time and reactivation time within perception. For each phase, the training time corresponding to that testing phase is highlighted in bold. This time period should always show a positive slope, indicating that classifiers trained and tested on time points belonging to the same phase are reactivated in the same order.

The slopes between the training time and the reactivation time of the different phases are shown below the hypothesised results in *Figure 5B*. As expected, the diagonal, reflecting training and testing on the same phase, was always positive. In contrast, training and testing on different phases tended to be associated with negative slopes. Note that for all decoding analyses, cross-validation was used, which means that these results cannot be due to overfitting but reflect true representational overlap between phases. To quantify the effect, we calculated a *Recurrence Index* (RI) which was defined as the dot-product between the vectorised hypothesis-matrix and the empirical-matrix. The RI is positive if the data show the hypothesised oscillatory pattern of slope reversals, zero if there is no clear oscillatory pattern and negative if the data show the opposite pattern. The RI was

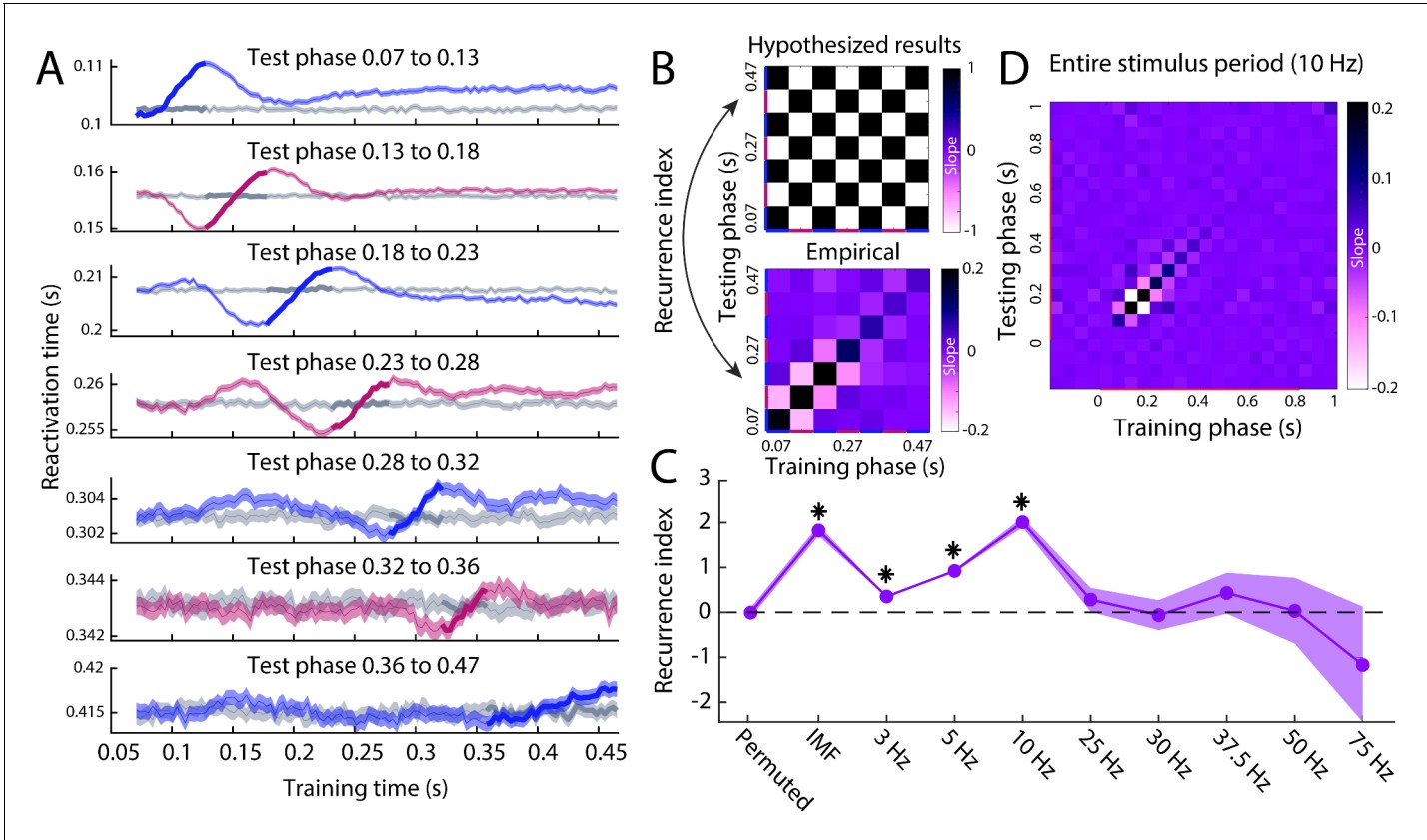

**Figure 5.** Reactivation timing for different perception phases. (A) The reactivation traces for each testing phase. Blue traces reflect feed-forward phases, pink traces reflect feedback phases (*Figure 4A*) and grey traces reflect reactivation traces for permuted classifiers. Shaded area represents the 95% confidence interval over trials. (B) Hypothesised (top) and empirical (bottom) slopes between the training and testing phases. The hypothesised matrix assumes recurrent processing such that subsequent phases show a reversal in the direction of information flow. Recurrence index reflects the amount of recurrent processing in the data, which is quantified as the dot product between the vectorised hypothesis matrix and empirical matrix. (C) Recurrence index for the permuted classifier, phase specification based on the IMF of the imagery reactivation trace (*Figure 4C*) and phase specification on evoked oscillations at various frequencies. (D) Slope matrix for phase specification defined at 10 Hz over the entire stimulus period. Source data associated with this figure can be found in the *Figure 5—source data 1–8*.

The online version of this article includes the following source data for figure 5:

**Source data 1.** Contains, for the segmentation of 3 Hz, Cfg = a configuration structure with the analysis options.
**Source data 2.** Contains, for the segmentation of 5 Hz, Cfg = a configuration structure with the analysis options.
**Source data 3.** Contains, for the segmentation of 10 Hz, Cfg = a configuration structure with the analysis options.
**Source data 4.** Contains, for the segmentation of 25 Hz, Cfg = a configuration structure with the analysis options.
**Source data 5.** Contains, for the segmentation of 30 Hz, Cfg = a configuration structure with the analysis options.
**Source data 6.** Contains, for the segmentation of 37.5 Hz, Cfg = a configuration structure with the analysis options.
**Source data 7.** Contains, for the segmentation of 50 Hz, Cfg = a configuration structure with the analysis options.
**Source data 8.** Contains, for the segmentation of 75 Hz, Cfg = a configuration structure with the analysis options.

significantly larger than zero for the true data (RI = 1.83, *CI* = 1.71 to 1.94, *p* < 0.0001) but not for the permuted data (RI = −0.008, *CI* = −0.12 to 0.10, p=0.548). This confirms that during perception, stimulus information flows up and down the visual hierarchy in feedback and feed-forward phases.

The phases that we used here were identified based on the IMF of the oscillation in the imagery reactivation trace (*Figure 4*), leading to phases of different lengths. Next, we investigated whether we could observe the same pattern if we specified the phases based on fixed evoked oscillations at different frequencies. For example, a 10 Hz oscillation resulted in four feed-forward and four feedback phases of 50 ms each within our 400 millisecond time window. The RI for the different frequencies is plotted in *Figure 5C*. Whereas the RI was significantly above zero for several low frequencies, the oscillatory pattern was clearest for the IMF based phases and for the 10 Hz oscillation,

confirming that the perceptual recurrence is most strongly aligned to the alpha frequency. Furthermore, to investigate whether this pattern of recurrence was indeed specific to the 400 millisecond time window identified previously, we also applied the 10 Hz recurrence analysis to the entire stimulus period (*Figure 5D*). The results show that the recurrence pattern is indeed restricted the first 400 ms after stimulus onset.

An interesting observation is that the recurrence pattern seems to be restricted to around the testing phase, such that only classifiers trained on phases close to the testing phase show a clear positive or negative relation with reactivation time. This is what causes the 'traveling wave' pattern between the rows in *Figure 5A*. An intriguing explanation for this observation is that stimulus representations change over subsequent cycles, such that representations only show a reactivation relation with neighbouring phases, but that the representations during later phases are too dissimilar to result in reliable reactivations. We tested whether recurrence was indeed specific to phases around the testing phase by comparing the normalised RI of slopes next to the diagonal in the slope matrix with the other slopes. Neighbouring phases indeed show a significantly higher RI (*M* = 0.073, *CI* = 0.069 to 0.076) than other phases (*M* = 0.002, *CI* = −0.0004 to 0.005, *p* < 0.0001), confirming that recurrence was restricted to a small number of cycles.

## Discussion

In this study, we investigated how stimulus representations flow through the visual system during mental imagery and perception. Our results reveal an asymmetry in information processing between perception and imagery. First, we showed that early perception processes are reactivated in reverse order during imagery, demonstrating that during imagery, activation flows from high-level visual areas down to low-level areas over time. Second, for later stages of perception, we found an oscillatory pattern of alternating positive and negative relations with imagery reactivation, indicating recurrent stimulus processing up and down the visual hierarchy aligned to an 11 Hz oscillation. Finally, by focusing on the identified feed-forward and feedback phases, we showed that recurrence during perception was restricted to neighbouring phases, suggesting that the format of neural stimulus representations changed with subsequent cycles of recurrent processing. Together, these findings indicate that during imagery, stimulus representations are activated via feedback processing whereas during perception, stimulus representations are iteratively updated through cycles of recurrent processing.

Our results are neatly in line with predictive processing (PP) theories. According to PP, recurrent processing during perception reflects dynamic hypothesis testing (*Bastos et al., 2012*; *Friston, 2005*; *Kersten et al., 2004*; *Knill and Pouget, 2004*). Specifically, perceptual inference is assumed to be accomplished via an interplay between top-down prediction signals encoding perceptual hypotheses and bottom-up prediction errors encoding the sensory signal unexplained by these hypotheses. Inferring the cause of sensory input is done by iteratively updating the perceptual hypothesis until the prediction error is minimised, in line with the dynamically changing representations observed here. Importantly, recurrent processing is assumed to happen hierarchically such that each level is activated by both bottom-up evidence as well as top-down predictions (*Friston, 2005*). This is in line with our observation that feed-forward and feedback sweeps proceeded at the same speed. Also in line with the current findings, PP predicts that these recurrent dynamics are dominated by slow-wave oscillations (*Bastos et al., 2012*). To our knowledge, the current study is the first to show these perceptual updating cycles empirically in humans. Furthermore, our results suggest that in the current task context, the perceptual inference process was completed in approximately four updating cycles. An exciting avenue for future research is to investigate whether the number of cycles needed can be modulated by task variables such as attention and stimulus noise.

Whereas perception was characterised by dynamically changing representations updated through recurrent cycles, we only found evidence for a single feedback flow during imagery. Specifically, all perceptual feed-forward sweeps showed a negative relationship with the same imagery time window and all perceptual feedback sweeps showed a negative relationship with that same imagery time window. This suggests that the imagery feedback flow contained the complete stimulus representation that was inferred via recurrent cycles during perception. This result fits with the idea that imagery uses the same predictive processes that underlie perceptual inference to run off-line simulations of sensory representations under different hypotheses (*Dijkstra et al., 2019*; *Gershman, 2019*;

*Grush, 2004*; *Hobson and Friston, 2012*; *Moulton and Kosslyn, 2009*). In contrast to perception, during imagery, there is no bottom-up sensory input prompting hypothesis updating. Instead, the perceptual cause is given and feedback connections are used to generate the corresponding low-level sensory representation based on the mapping that was learned during perception. Recurrent processing within the visual system might become important when imagining a more dynamic stimulus in which sensory representations change over time. However, it is also possible that recurrent dynamics were actually present during imagery in this task but that we were unable to reveal them due to signal-to-noise issues. Future research should focus on developing more sensitive techniques to further characterise information flow during imagery. Another interesting question for future research is whether reactivation during mental imagery has to always fully progress down the visual hierarchy. In this study, participants were instructed to generate highly detailed mental images and catch trials were used to ensure that they indeed focused on low-level visual details. It might be the case that if less detail is needed for the task, earlier perception processes are not reactivated and mental simulations stop at a higher level (*Kosslyn and Thompson, 2003*; *Pearson and Keogh, 2019*).

Central to this predictive processing interpretation of bottom-up and top-down sweeps is that increasingly abstract stimulus features are processed in higher-level brain areas. This is indeed what has generally been observed in the literature (*Hubel and Wiesel, 1968*; *Thorpe and Fabre-Thorpe, 2001*; *Vogels and Orban, 1996*). However, because we have not directly assessed which stimulus features were captured by the classifiers in this study, we cannot be certain that the sweeps through the visual hierarchy observed here genuinely reflect processing of stimulus information at different levels of abstraction. For example, it is possible for the classifier to be driven by other features of the signal irrelevant to processing of visual information such as general amplification of signal due to differences in attentional capture between the two stimuli, which might have also given rise to sweeps through the visual hierarchy (*Michalareas et al., 2016*). It is likely that such high-level cognitive mechanisms have influenced processing during later stages of perception. However, a recent study showed that the initial feedforward sweep during perception does not seem to be influenced by attention (*Alilović et al., 2019*). Moreover, modelling work using Convolutional Neural Networks (CCNs) has shown that the MEG signal during perception does reflect activation of hierarchically increasing complex features over time (*Seeliger et al., 2018*). Further research using explicit encoding models of stimulus features at different levels of abstraction would be necessary to completely address this point. Furthermore, the current study used a number of non-traditional analysis steps. While we aimed to demonstrate the validity of this approach via simulations, it is worth noting that simulations are not a perfect control since simulated data cannot account for all the features present in real data, and might be blind to other issues. Therefore, to fully ensure that this analysis approach does not suffer from any overlooked confounds, future validation studies are needed.

In conclusion, by using a novel multivariate decoding approach which allowed us to infer the order in which representations were reactivated, we show that, while similar neural representations are activated during imagery and perception, the neural dynamics underlying this activation are different. Whereas perception is characterised by recurrent processing, imagery is dominated by top-down feedback processing. These results are in line with the idea that during perception, high-level causes of sensory input are inferred whereas during imagery, this inferred mapping is reversed to generate sensory representations given these causes. This highlights a fundamental asymmetry in information processing between perception and imagery and sets the stage for exciting new avenues for future research.

## Materials and methods

### Participants

We assumed a medium effect size (d = 0.6) which, to reach a power of 0.8, required twenty four participants. To take into account drop-out, thirty human volunteers with normal or corrected-to-normal vision gave written informed consent and participated in the study. Five participants were excluded: two because of movement in the scanner (movement exceeded 15 mm), two due to incorrect execution of the task (less than 50% correct on the catch trials, as described below) and one due to technical problems. Twenty-five participants (mean age 28.6, *SD* = 7.62) remained for the final analysis.

The study was approved by the local ethics committee and conducted according to the corresponding ethical guidelines (CMO Arnhem-Nijmegen). An initial analysis of these data has been published previously (*Dijkstra et al., 2018*).

## Experimental design

We adapted a retro-cue paradigm in which the cue was orthogonalised with respect to the stimulus identity (*Harrison and Tong, 2009*). Participants were shown two images after each other followed by a retro-cue indicating which of the images had to be imagined. After the cue, a frame was shown in which the participants had to imagine the cued stimulus as vividly as possible. Next, they had to indicate their experienced imagery vividness by moving a bar on a continuous scale. To ensure that participants were imagining the stimuli with great visual detail, both categories contained eight exemplars, and on 7% of the trials the participants had to indicate which of four exemplars they imagined. The exemplars were chosen to be highly similar in terms of low-level features to minimise within-class variability and increase between-class classification performance. We instructed participants to focus on vividness and not on correctness of the stimulus, to motivate them to generate a mental image including all visual features of the stimulus. The stimuli encompassed $2.7 \times 2.7$ visual degrees. A fixation bull's-eye with a diameter of 0.1 visual degree remained on screen throughout the trial, except during the vividness rating.

## MEG recording and preprocessing

Data were recorded at 1200 Hz using a 275-channel MEG system with axial gradiometers (VSM/CTF Systems, Coquitlam, BC, Canada). For technical reasons, data from five sensors (MRF66, MLC11, MLC32, MLF62, MLO33) were not recorded. Subjects were seated upright in a magnetically shielded room. Head position was measured using three coils: one in each ear and one on the nasion. Throughout the experiment head motion was monitored using a real-time head localiser (*Stolk et al., 2013*). If necessary, the experimenter instructed the participant back to the initial head position during the breaks. This way, head movement was kept below 8 mm in most participants. Furthermore, both horizontal and vertical electro-oculograms (EOGs), as well as an electrocardiogram (ECG) were recorded for subsequent offline removal of eye- and heart-related artefacts. Eye position and pupil size were also measured for control analyses using an Eye Link 1000 Eye tracker (SR Research). Data were analysed with MATLAB version R2018a and FieldTrip (*Oostenveld et al., 2011*) (RRID:SCR_004849). Per trial, three events were defined. The first event was defined as 200 ms prior to onset of the first image until 200 ms after the offset of the first image. The second event was defined similarly for the second image. Further analyses focused only on the first event, because the neural response to the second image is contaminated by the neural response to the first image. Finally, the third event was defined as 200 ms prior to the onset of the retro-cue until 500 ms after the offset of the imagery frame. As a baseline correction, for each event, the activity during 300 ms from the onset of the initial fixation of that trial was averaged per channel and subtracted from the corresponding signals. The data were down-sampled to 300 Hz to reduce memory and CPU load. Line noise at 50 Hz was removed from the data using a DFT notch filter. To identify artefacts, the variance of each trial was calculated. Trials with high variance were visually inspected and removed if they contained excessive artefacts. After artefact rejection, on average 108 perception face trials, 107 perception house trials, 105 imagery face trials and 106 imagery house trials remained for analysis. To remove eye movement and heart rate artefacts, independent components of the MEG data were calculated and correlated with the EOG and ECG signals. Components with high correlations were manually inspected before removal. The eye tracker data were cleaned separately by inspecting trials with high variance and removing them if they contained blinks or other excessive artefacts.

## Reactivation timing analysis

To estimate when neural representations at different perception time points got reactivated during imagery, we trained classifiers on several time points during perception and applied them to imagery. Specifically, to decode the stimulus category per perception time point, we used a linear discriminant analysis (LDA) classifier with the activity from the 270 MEG sensors as features and a shrinkage regularisation parameter of 0.05 (see [*Mostert et al., 2016*] for more details). To prevent a potential bias in the classifier, the number of trials per class was balanced per fold by randomly

removing trials from the class with the most trials until the trial numbers were equal between the classes. In our design, the perception and imagery epochs happened in the same general 'trial'. If the imagery epochs on which the classifier was tested came from the same trials as the perception epochs on which it was trained, auto-correlations in the signal could inflate decoding accuracy. To circumvent this, a five-fold cross-validation procedure was implemented where for each fold the classifier was trained on 80% of the trials and tested on the other 20%.

Because the amplitude of the signal is usually smaller during imagery than during perception (*Kosslyn et al., 2001*; *Pearson and Keogh, 2019*), we demeaned the data by subtracting the mean over trials per time point and per sensor. This forced the classifier to leverage relative changes in the multivariate signal while avoiding the confounding effect of having global changes in amplitude.

This resulted in a decision value for each perception model, for each imagery trial and time point. The sign of this value indicates in which category the trial has been classified and the value indicates the distance to the hyperplane. In order to obtain the confidence of the classifier in the correct class, the sign of the distance values in one category was inverted. This means that increasing positive distance values now always reflected increasingly confident classification. To obtain the specific moment within each imagery trial that a given perception model became active, we identified the time point with the highest distance (*Linde-Domingo et al., 2019*; *Figure 1*).

Identifying the order of reactivation of neural representations was then done by performing a linear regression with the training time points as predictor and the inferred reactivation times points as dependent variable. A positive relationship indicates that reactivations happened in the same order, whereas a negative relationship indicates a reversal in the order of reactivations.

After discovering the oscillatory pattern between later perception time points and imagery reactivation, we also ran this reactivation analysis within perception by training on all perception time points and testing on specific perception time windows reflecting the identified feed-forward and feedback phases (*Figure 4*). This identified, per phase, reversals in information flow.

For the imagery generalisation, the time window used to obtain the peak distance extended from the cue onset until the vividness instruction onset, covering the entire 4 s during which participants were instructed to imagine the stimulus. For this data, we removed high frequency noise using a low-pass filter of 30 Hz (for the results using the raw data, see *Figure 3—figure supplement 1*). For the within-perception tests we did not use a low-pass filter because we used smaller testing time windows and were also interested in possible high-frequency effects.

To ensure that the observed effects were due to dynamics in activation of neural stimulus representations, and not due to dynamics of the raw signals, for each analysis, we performed the same analysis after permuting the class-labels, thereby removing stimulus information from the data without altering the temporal structure of the data.

## Simulations

We tested the validity of our approach on a relatively realistic synthetic dataset. Specifically, we tested whether the trial-by-trial LDA distance measures could successfully be used to infer the order of reactivation of neural representations, even when the timing of these reactivations differed between trials. The true neural model consisted of 5 neural representations activated in a partially overlapping sequence over the course of 60 milliseconds, with a sampling rate of 300 Hz, modelled as pseudo random activation of 20 sensors for two classes (*Figure 2A*). Stimulus information is defined as the difference between the two classes. The training set was generated as 100 trials per class of the true neural model activation. Testing sets were generated as 100 trials per class of the true neural model activation in the same order or in reversed order as the training set and slowed down by a factor of 10. Furthermore, temporal uncertainty between trials was introduced by randomly sampling the onset of the reactivation in each trial from a standard normal distribution with mean 0.8 s and a standard deviation of 0.1 or 0.5. Testing set trials were 2 s long with a sampling frequency of 300 Hz. This stimulus-specific activity was added on top of some generated ongoing activity, which we simulated as realistic 1/f noise. In order to do that, we fitted a multivariate autoregressive model of order 1 to the real data, and then sampled from it trial by trial (*Vidaurre et al., 2019*). We simulated 10 subjects and performed LDA cross-decoding analysis as described above in each subject separately resulting in distance measures per testing trial and time point. Decoding accuracy was computed for each combination of training and testing time point as the number of trials in which the distance was highest for the correct class divided by the total number of trials (i.e.

proportion of correctly classified trials). Reactivation time was computed for each training time point and for each trial as the testing time with the peak distance in favour of the correct class (*Linde-Domingo et al., 2019*).

### Realignment

To confirm that the order of activation of low and high-visual areas was reversed during imagery compared to perception, we realigned the imagery activation based on the identified distance peaks. This was done by selecting the peak imagery time point for every trial for every perception model time point to create a realigned imagery data set. The time axis for this new data set was inferred using the linear relationship between perception model time and imagery reactivation time established in the main imagery reactivation analysis. Source activation was then calculated using the same procedure as was used for the un-realigned perception and imagery data (more details below under Source localisation).

### Frequency analysis

For the time frequency analysis, we used a Morlet Wavelet at 10 Hz defined as:

$$MW = \exp(\frac{-t^2}{2s^2} + iwt)$$

where $t$ is a time vector from $-0.5$ to $0.5$ in steps of 1/fs, $w = 2\pi 10$, $s = \frac{c2\pi}{w}$ and $c$ is the number of cycles, this case 1. We only used 200 samples in the centre of the wavelet and convolved this with the mean reactivation trace to obtain the time frequency representation. To calculate power at different frequencies we used the Fast Fourier Transform (FFT). To prevent edge effects, we first multiplied the mean signal with a Hanning taper from $-0.2$ to 0.6 s prior to performing the FFT. Of the resulting complex numbers, the absolute value was taken and the result was normalised by the length of the signal.

Within reactivation traces, positive slopes represented reactivation in a similar order whereas negative slopes represented reactivation in reverse order. We wanted to divide the signal into phases of positive and negative slopes because these represented feed-forward and feedback phases. In order to do this, we used empirical mode decomposition (EMD) which separates the signal into intrinsic mode functions (IMF) based on local and global extrema; that is, peaks and troughs (*Huang et al., 1998*; *Rilling et al., 2003*; *Wang et al., 2010*). This technique identifies oscillations without assuming that these oscillations should be sinusoidal. For a 10 Hz frequency we would expect eight extrema in 400 ms, reflecting four full cycles. Therefore, to identify the different phases, we selected the IMF with the number of extrema closest to eight. Decreasing and increasing phases were defined as periods between subsequent extrema (*Figure 4C*). Slopes between periods were calculated using linear regression and slopes for decreasing and increasing periods were averaged to reflect the speed of feed-forward and feedback processing respectively. To determine the uncertainty of these slopes, for every bootstrapping sample the mean reactivation trace and the corresponding EMD separation was recalculated.

### Statistics

To test whether there was a significant linear relationship between perception training model time and imagery reactivation time, we used a generalised linear mixed model (GLMM) with the single trial classifier distance peaks as dependent variable and perception model time during the feed-forward sweep as independent variable. We chose GLMMs because they make fewer assumptions than more commonly used GLMs and because we expected large differences in the onset of reactivation between trials and subjects and, in contrast to GLMs, GLMMs allow for random effects on trials and subjects.

To obtain 95% confidence intervals for reactivation times, time-frequency and frequency plots, we performed bootstrapping analyses with 10000 bootstrapping samples. For pair-wise comparisons, we obtained p-values by bootstrapping the difference between the two conditions. Source traces represented the mean difference between the stimulus classes which cannot be computed per trial. Therefore, uncertainty in the mean of these values was represented as the standard error of the mean (SEM) over subjects.

## Source localisation

To identify brain areas that represented information about the stimuli during perception and imagery, we performed source reconstruction. For LDA classification, the spatial pattern that underlies the classification (i.e. the decoding weights or 'stimulus information'), reduces to the difference in magnetic fields between the two conditions (*Haufe et al., 2014*). Therefore, the difference ERF between faces and houses reflects the contributing brain areas. For the sensor-level activation plots, we calculated the planar gradient for each participant prior to averaging over participants. For the source-level plots, we performed source reconstruction on the axial difference ERF.

T1-weighted structural MRI images were acquired in separate sessions using a Siemens 3T MRI scanner. Vitamin E markers in both ears indicated the location of the head coils during the MEG measurements, allowing for realignment between the two. The location of the fiducial at the nasion was estimated based on anatomy. The volume conduction model was created based on a single shell model of the inner surface of the skull. The source model was based on a reconstruction of the cortical surface created for each participant using FreeSurfer's anatomical volumetric processing pipeline (RRID:SCR_001847,(*Fischl, 2012*). MNE-suite (Version 2.7.0; https://mne.tools/, RRID:SCR_005972, (*Gramfort et al., 2014*) was subsequently used to infer the subject-specific source locations from the surface reconstruction. The resulting head model and source locations were co-registered to the MEG sensors.

The lead fields were rank reduced for each grid point by removing the sensitivity to the direction perpendicular to the surface of the volume conduction model. Source activity was obtained by estimating linearly constrained minimum variance (LCMV) spatial filters (*Van Veen et al., 1997*). The data covariance was calculated over the interval of 50 ms to 1 s after stimulus onset for perception and over the entire segment for imagery. The data covariance was subsequently regularised using shrinkage with a regularisation parameter of 0.01 (as described in *Manahova et al., 2018*). These filters were then applied to the sensor MEG data, resulting in an estimated two-dimensional dipole moment for each grid point over time.

To facilitate interpretation and visualisation, we reduced the two-dimensional dipole moments to a scalar value by taking the norm of the vector. This value reflects the degree to which a given source location contributes to activity measured at the sensor level. However, the norm is always a positive value and will therefore, due to noise, suffer from a positivity bias. To counter this bias, we employed a permutation procedure in order to estimate this bias. Specifically, in each permutation, the sign of half of the trials were flipped before averaging and projecting to source space. This way, we cancelled out the systematic stimulus-related part of the signal, leaving only the noise. Reducing this value by taking the norm thus provides an estimate of the positivity bias. This procedure was repeated 1000 times, resulting in a distribution of the noise. We took the mean of this distribution as providing the most likely estimate of the noise and subtracted this from the true, squared source signal. Furthermore, this estimate provides a direct estimate of the artificial amplification factor due to the depth bias. Hence, we also divided the data by the noise estimate to obtain a quantity that allowed visualisation across cortical topography, leading to an unbiased estimate of the amount of stimulus information present in each cortical area. Values below zero therefore reflected no detectable signal compared to noise. For full details, see *Manahova et al., 2018*.

To perform group averaging, for each subject, the surface-based source points were divided into 74 atlas regions as extracted by FreeSurfer on the basis of the subject-specific anatomy (*Destrieux et al., 2010*). Next, the activation per atlas region was averaged over grid points for each participant. Group-level activations were then calculated by averaging the activity over participants per atlas region (*van de Nieuwenhuijzen et al., 2016*). The early visual cortex ROI (EVC) corresponded to the 'occipital pole' parcels from the Destrieux atlas and the inferior temporal ROI (IT) was a combination of the 'temporal lateral fusiform', 'temporal lateral' and 'temporal lateral and lingual' parcels. Activation in the IT ROI was calculated by applying PCA to the three parcels and taking the first principal component. Because data are z-scored over time during PCA, to ensure that the activation in the EVC ROI was comparable to activation in IT, we also z-scored these data.

## Acknowledgements

The authors would like to thank Emma K Ward for help with the linear mixed model statistics. This work is supported by VIDI grant (639.072.513) from the Netherlands Organization for Scientific Research.

## Additional information

### Funding

| Funder | Grant reference number | Author |
|---|---|---|
| Netherlands Organisation for Scientific Research | 639.072.51 | Luca Ambrogioni<br>Marcel van Gerven |

The funders had no role in study design, data collection and interpretation, or the decision to submit the work for publication.

### Author contributions

Nadine Dijkstra, Conceptualization, Data curation, Formal analysis, Funding acquisition, Investigation, Visualization, Methodology, Writing - original draft, Project administration, Writing - review and editing; Luca Ambrogioni, Conceptualization, Formal analysis, Visualization, Writing - original draft, Writing - review and editing; Diego Vidaurre, Conceptualization, Formal analysis, Supervision, Writing - review and editing; Marcel van Gerven, Resources, Supervision, Funding acquisition, Writing - review and editing

### Author ORCIDs

Nadine Dijkstra (iD) https://orcid.org/0000-0003-1423-9277
Diego Vidaurre (iD) http://orcid.org/0000-0002-9650-2229
Marcel van Gerven (iD) https://orcid.org/0000-0002-2206-9098

### Ethics

Human subjects: All participants gave written informed consent. The study was approved by the local ethics committee and conducted according to the corresponding ethical guidelines (CMO Arnhem-Nijmegen; ECSW2014-0109-246a van Gerven).

### Decision letter and Author response

Decision letter https://doi.org/10.7554/eLife.53588.sa1
Author response https://doi.org/10.7554/eLife.53588.sa2

## Additional files

### Supplementary files

• Supplementary file 1. Linear Mixed Model Bayesian Information Criterion (BIC) output for the different models. (a) Bayesian Information Criterion (BIC) for linear mixed-effects model explaining imagery reactivation time with perception model time. (b) BIC for linear mixed-effects model explaining imagery reactivation time with perception model time after permuting the stimulus-class labels. (c) BIC for linear mixed-effects model explaining imagery reactivation time averaged over trials within subject with perception model time.

• Transparent reporting form

### Data availability

The data used in this paper is available at http://hdl.handle.net/11633/di.dcc.DSC_2017.00072_245 and the analysis code is available at https://github.com/NadineDijkstra/IMAREV.git (copy archived at https://github.com/elifesciences-publications/IMAREV).

The following previously published dataset was used:

| Author(s) | Year | Dataset title | Dataset URL | Database and Identifier |
|-----------|------|---------------|-------------|-------------------------|
| Dijkstra N, Mostert P, de Lange FP, Bosch SE, van Gerven MAJ | 2018 | Temporal dynamics of visual imagery and perception | http://hdl.handle.net/11633/di.dcc.DSC_2017.00072_245 | Donders Depository DSC_2017.00072_245:v1, 245:v1 |

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
