## [Decision Letter]

**Acceptance summary:**

Dijkstra et al. provide empirical evidence for reversal of perceptual inference by analyzing the electrophysiological signature of brain responses to visual stimuli. It reveals a feedback loop from higher-level regions that appear as an oscillation in the delay of the brain response. This empirical evidence is useful to refine theories of perception.

**Decision letter after peer review:**

Thank you for submitting your article "Neural dynamics of perceptual inference and its reversal during imagery" for consideration by *eLife*. Your article has been reviewed by three peer reviewers, one of whom is a member of our Board of Reviewing Editors, and the evaluation has been overseen by Timothy Behrens as the Senior Editor. The reviewers have opted to remain anonymous.

The reviewers have discussed the reviews with one another and the Reviewing Editor has drafted this decision to help you prepare a revised submission.

Summary:

This manuscript uses MEG during a working memory/imagery task to investigate the flow of information between low and high level areas during perception and imagery respectively. The authors provide empirical evidence consistent with the notion that high level regions are first during imagery, and last during perception. In addition, they show that recall leads to several oscillations (around 11Hz), interpreted as bottom-up and top-down processes. This evidence is extracted via several methodological innovations. A multivariate classification methods, namely Linear Discriminant Analysis (LDA) is applied to various time points following stimulus presentation and the resulting trained classifiers (one by time-point) are used, to time-align the imagery trials via their ability to discriminate better than others. This time alignment is central to claim that “information” is processed bottom-up or top-down along the ventral visual pathway. Here the distance to the discriminating hyperplane is used as a proxy for the evidence that the signal is present, analyzed at the group level.

The findings were deemed of high interest by the reviewers. The evidence for oscillations is expected to stimulate greatly neurosciences, in particular predictive coding theory. Indeed, the phenomena revealed are insufficiently known (e.g. the oscillation between feedforward and feedback activity during imagery). The reviewers also appreciated the creative use of classifiers to extract signal as well as the fact that the data from the study are publicly available.

The discussions however mostly focused on trying to decide whether or not the evidence was conclusive. Given the amount of non-classic data transformations that lead to the evidence, how to assert that the results, in particular the multiple oscillations, are specific to visual recall, and not low-level properties of the neural signal, Indeed, the neural signal has a complex time-frequency structure. Our understanding of the permutations used is that they will create unstructured patterns, which will lead to a random and unstructured ordering of the most-similar pattern. In other terms, the permutations do not capture the temporal structure of the neural signal. A better null procedure is needed to provide solid evidence that the oscillation revealed is indeed related to imagery dynamics.

Essential revisions:

1) A first point to clarify are what aspects of the stimuli that drives variations in the perception time. The manuscript is very light on details in terms of the psychological dimensions varied to create the variations in the perception time that are related to variations in the imagery time. The theories invoked (predictive processing theories) relate to ambiguity in the sensory signal, hence they suggest that the variations the perception time should be related to the stimuli. Yet, the manuscript does not offer an explanation on the causes the variation in perception time. Some of the differences between the two classes of stimuli likely reflect "unspecific" mechanisms: i.e. electrophysiological signatures that are not specific to particular neural representations, but to overall differences in e.g. attentional capture. For example, if faces trigger a stronger visual response than houses, then, what is considered to be the reactivation of a face in this study, would simply reflect an overall modulation of visual/IT activity. Such non-specific activation would be expected to trigger feedforward and feedback traveling waves in the ventral and dorsal visual pathways, as extensively described in monkey electrophysiology as well as with MEG (e.g. Michalareas et al., 2016). For these reasons, we feel that the revised manuscript should discuss in detail what aspects of the stimuli drive the variations in time.

2) A second wider point relates to establishing whether or not the sophisticated analysis enhances oscillations that are specific to mental imagery, or rather general to neural signals. Indeed, neural signals have a complex autocorrelation structure. To address the possible confound that alpha modulation could affect the SNR and explain the corresponding findings the authors perform a coherence analysis on individual sensors. The fact that no significant effect is obtained with this analysis motivates the authors to rule out this hypothesis. However, statistics on single sensors are known to be less statistically powerful techniques, so the result could be explained by the poor statistical power of this supplementary analysis. Overall, all the reviewers found it hard to assess the methods, and hence feel that the whole pipeline should be experimentally shown not to create the signatures that are interpreted. A small number of specific analyses can help establishing this evidence without resorting to fine theoretical analysis of the data-processing pipeline:

A) Applying the same exact analysis to neural signal where one would not expect such conclusion to arise. A specific instance of such signals can be found using time windows located in the inter-stimuli intervals (for instance before the trial).

B) Using a complementary analysis based on second-level statistics: i.e. fit a distinct model on each subject separately, and test the reactivation signatures across subjects. Indeed, the authors use mixed-model/first-level statistics with trials and subjects as random variables. Such approach is unusual in decoding-based analyses, because of the cross-validation – i.e. the training of the decoder introduces dependencies across decoders' predictions, and thus breaks the independence assumption of first-order statistical tests.

C) Using the non low-pass filtered data. Indeed, there is the suspicion that the oscillatory effect observed during perception could be due to a low pass filtering effect. This doubt is suggested by the supplementary analysis using non-low pass filtered data. Figure S2 in supplementary material is however not comparable to the corresponding figure in the main text. If the non-low pass filtered data actually confirm the findings these data should be used in the main text without the need for added analysis.

[Editors' note: further revisions were suggested prior to acceptance, as described below.]

Thank you for resubmitting your work entitled "Neural dynamics of perceptual inference and its reversal during imagery" for further consideration by *eLife*. Your revised article has been evaluated by Timothy Behrens (Senior Editor) and a Reviewing Editor.

The manuscript has been improved but there are some remaining issues that need to be addressed before acceptance, as outlined below:

The reviewers find that the evidence is not rock solid, but the work is stimulating and the findings do not stem from an apparent methodological flaw. They feel that publishing these is beneficial for the field, but would like the potential limitations to be more explicit. No additional analysis is mandatory for resubmitting, but work on the wording is important.

Reviewer #1:

The authors answered thoroughly most of the comments raised by the reviewers. Importantly, they performed complementary analyses to make the evidence stronger, for instance the analysis without low-pass filtering. Also, the trial-level data show a weak effect, but one that is present, which is an important piece of evidence to confirm that the effect is not created by the analysis. They unfortunately did not follow our suggestion of analyzing data outside the stimuli, which would have made a strong null hypothesis. Rather, they used an elaborate simulation.

Overall, I feel comforted that the evidence is not created by the data analysis, and I am in favor of publication.

Reviewer #2:

Thanks to the authors for this response.

Generally speaking, I am not amazed by responses based on simulations – which is the source of a significant set of the new results and analyses in the present revision. Simulations remain soft controls, because, unlike real data, they may be blind to a wide variety of issues. Responses systematically based on novel analyses of the MEG signals would have been more convincing, like we originally proposed (e.g. applying the same analyses on different time windows such as the inter-stimulus interval, in order to ensure that no effect were found there).

In addition, if I understand correctly, the p-values derived from the mix models may be invalid (reviewing comment 2B): i.e. we cannot easily estimate p-values when the independence hypotheses between samples is not met – which is the case with a CV, because the training models are fitted on partially similar datasets – consequently their predictions are not independent: c.f. e.g. Noirhomme et al., 2014 (NeuroImage: Clinical).

This arguably minor concern is reinforced by the fact that single-trial-based statistics can dramatically increase the degrees of freedom, and can thus lead to unreasonably confident p-values. Second-level stats across subjects only would be a more conservative approach. Again, the authors did not follow that recommended route.

That being said, these statistical issues should not undermine the methodological and neuroscientific work. While I am relatively unhappy with the current answer, the paper remains interesting and potentially important.

Consequently, I will not oppose its acceptation for publication, but would strongly recommend the authors to provide a clear code of their analyses and simulations to facilitate future replications.

Reviewer #3:

I would like to thank the authors for the convincing and thorough revision.

I just have two remaining comments.

– Related to your answer "We acknowledge that this breaks the independence assumption but we still believe that this is the most valid test for our data, since testing across subjects ignores the large between-trial variability." Can you add a sentence somewhere in the text about this?

Just as a comment here, using non-parametric methods like in the EEGLAB LIMO toolbox would address this issue: e.g. Pernet et al., 2011 (Computational Intelligence and Neuroscience).

– The proper URL for MNE software is https://mne.tools/ and the related

publication is:

A. Gramfort, M. Luessi, E. Larson, D. Engemann, D. Strohmeier, C. Brodbeck, L. Parkkonen, M. Hämäläinen, MNE software for processing MEG and EEG data, NeuroImage, Volume 86, 1 February 2014, Pages 446-460, ISSN 1053-8119

---

## [Author Response]

Essential revisions:1) A first point to clarify are what aspects of the stimuli that drives variations in the perception time. The manuscript is very light on details in terms of the psychological dimensions varied to create the variations in the perception time that are related to variations in the imagery time. The theories invoked (predictive processing theories) relate to ambiguity in the sensory signal, hence they suggest that the variations the perception time should be related to the stimuli. Yet, the manuscript does not offer an explanation on the causes the variation in perception time. Some of the differences between the two classes of stimuli likely reflect "unspecific" mechanisms: i.e. electrophysiological signatures that are not specific to particular neural representations, but to overall differences in e.g. attentional capture. For example, if faces trigger a stronger visual response than houses, then, what is considered to be the reactivation of a face in this study, would simply reflect an overall modulation of visual/IT activity. Such non-specific activation would be expected to trigger feedforward and feedback traveling waves in the ventral and dorsal visual pathways, as extensively described in monkey electrophysiology as well as with MEG (e.g. Michalareas et al., 2016). For these reasons, we feel that the revised manuscript should discuss in detail what aspects of the stimuli drive the variations in time.

We thank the reviewers for raising this important point. We agree that it is important to discuss in more detail the stimulus features that can be captured by our classifiers during perception and subsequently during the reactivation during imagery.

The main analysis is based on the idea that stimulus features of increasing complexity are processed in a hierarchical manner during the initial time points of perception (i.e. “the feedforward sweep”) and that this is what is leveraged by our classifiers. We confirmed that early classifiers indeed picked up differences in neural activation in low-level visual areas whereas later classifiers relied on differences in high-level visual areas using source-reconstruction (Figure 1A-B). Therefore, they support our main claims, which use these classifiers to investigate whether stimulus information – defined as differences in neural activation between the two stimuli – flows up or down the visual hierarchy.

However, as the reviewers point out, the inference that these classifiers pick up on increasingly complex stimulus features is based on what the literature says about the features that are processed in these different brain areas. We did not directly assess which stimulus features the classifier was sensitive to in this experiment and therefore cannot be certain about which features each classifier decoded. To discuss this point in more detail we have added the following paragraph to the Discussion:

“Central to this predictive processing interpretation of bottom-up and top-down sweeps is that increasingly abstract stimulus features are processed in higher-level brain areas. […] Further research using explicit encoding models of stimulus features at different levels of abstraction would be necessary to completely address this point.”

2) A second wider point relates to establishing whether or not the sophisticated analysis enhances oscillations that are specific to mental imagery, or rather general to neural signals. Indeed, neural signals have a complex autocorrelation structure. To address the possible confound that alpha modulation could affect the SNR and explain the corresponding findings the authors perform a coherence analysis on individual sensors. The fact that no significant effect is obtained with this analysis motivates the authors to rule out this hypothesis. However, statistics on single sensors are known to be less statistically powerful techniques, so the result could be explained by the poor statistical power of this supplementary analysis. Overall, all the reviewers found it hard to assess the methods, and hence feel that the whole pipeline should be experimentally shown not to create the signatures that are interpreted. A small number of specific analyses can help establishing this evidence without resorting to fine theoretical analysis of the data-processing pipeline:

We agree with the reviewers that the method pipeline is unorthodox and needs to be validated more thoroughly. In order to do this, we ran a simulation, added details on the relationship between our approach and well-established decoding and added the results using the unfiltered data.

With respect to the relationship between the observed oscillation in reactivations of neural representations and alpha power, we agree that statistics on single sensors are less statistically sensitive. However, the observed coherence is so low and has such a random topography that it is unlikely that the lack of a finding here is only due to statistical power. To clarify this, we have added the coherence topoplot as a sub-panel of the main oscillation figure, see Figure 4D.

A) Applying the same exact analysis to neural signal where one would not expect such conclusion to arise. A specific instance of such signals can be found using time windows located in the inter-stimuli intervals (for instance before the trial).

We agree with the reviewer that this is a good way of testing the validity of our approach. However, we have decided to focus on showing the validity of our approach using simulations, which allowed us to directly test the effect of different parameters (e.g. order of reactivation and amount of temporal uncertainty between trials) on the observed results.

B) Using a complementary analysis based on second-level statistics: i.e. fit a distinct model on each subject separately, and test the reactivation signatures across subjects. Indeed, the authors use mixed-model/first-level statistics with trials and subjects as random variables. Such approach is unusual in decoding-based analyses, because of the cross-validation – i.e. the training of the decoder introduce dependencies across decoders' predictions, and thus breaks the independence assumption of first-order statistical tests.

We are not completely sure that we understand what the reviewers refer to here. For clarification, we do fit a distinct perception decoding model to each subject separately and calculate the reactivation of the subject-specific model per trial during imagery. Because we expected large variation in onset between trials, we applied a Linear Mixed Model which allowed us to set “trial” as a random variable. This is equivalent to what was used in Linde-Domingo et al., 2019, on which we based a large part of our methods. We did use cross-validation because the perception and imagery events were part of the same “trial” period, resulting in auto-correlation between imagery and perception if they belonged to the same trial (see Materials and methods “Temporal Decoding Analysis”). Is the reviewer referring to the fact that there is some dependency across predictions within each subject because the training data for different folds overlaps? We acknowledge that this breaks the independence assumption but we still believe that this is the most valid test for our data, since testing across subjects ignores the large between-trial variability.

C) Using the non low-pass filtered data. Indeed, there is the suspicion that the oscillatory effect observed during perception could be due to a low pass filtering effect. This doubt is suggested by the supplementary analysis using non-low pass filtered data. Figure S2 in supplementary material is however not comparable to the corresponding figure in the main text. If the non-low pass filtered data actually confirm the findings these data should be used in the main text without the need for added analysis.

The low-pass filter was only applied to the imagery distance values and not to the perception data, and therefore could not have induced an oscillation during perception. To clarify this further we have added the low-pass filter step to the methods figure, see Figure 1.

Furthermore, we agree that the supplementary figures showing the unfiltered results were not clear before as they used a different y-axis. We have now added the results without this filtering step as supplementary figures to both Figure 2 and Figure 3.

As can be seen, none of the results qualitatively changed when we removed the filter. However, we decided to use the filter because during imagery there is a lot of uninformative high-frequency signal, purely because the signal is quite noisy. This is apparent in the fact that the results become cleaner with the filter, especially for the first imagery reversal. Because the filter could not induce the observed oscillation and because the results became cleaner, we decided to keep the filtered results as the main figures in the paper.

[Editors' note: further revisions were suggested prior to acceptance, as described below.]

The manuscript has been improved but there are some remaining issues that need to be addressed before acceptance, as outlined below:The reviewers find that the evidence is not rock solid, but the work is stimulating and the findings do not stem from an apparent methodological flaw. They feel that publishing these is beneficial for the field, but would like the potential limitations to be more explicit. No additional analysis is mandatory for resubmitting, but work on the wording is important.

Thanks. Even though no further analyses were mandatory, we have nevertheless preferred to address the remaining concern regarding the validity of our statistical tests by adding another analysis in which we test between subjects rather than between trials and show that the effect remains significant. Furthermore, we added a paragraph on the limitations of the current analysis pipeline and have made all analysis scripts publicly available on GitHub. We hope that this has sufficiently addressed the raised concerns.

Reviewer #2:Thanks to the authors for this response.Generally speaking, I am not amazed by responses based on simulations – which is the source of a significant set of the new results and analyses in the present revision. Simulations remain soft controls, because, unlike real data, they may be blind to a wide variety of issues. Responses systematically based on novel analyses of the MEG signals would have been more convincing, like we originally proposed (e.g. applying the same analyses on different time windows such as the inter-stimulus interval, in order to ensure that no effect were found there).

We understand the reviewer’s point about the potential issues with simulations. To address this, we have added the following to the Discussion:

“Furthermore, the current study used a number of non-traditional analysis steps. While we aimed to demonstrate the validity of this approach via simulations, it is worth noting that simulations are not a perfect control since simulated data cannot account for all the features present in real data, and might be blind to other issues. Therefore, to fully ensure that this analysis approach does not suffer from any overlooked confounds, future validation studies are needed.”

In addition, if I understand correctly, the p-values derived from the mix models may be invalid (reviewing comment 2B): i.e. we cannot easily estimate p-values when the independence hypotheses between samples is not met – which is the case with a CV, because the training models are fitted on partially similar datasets – consequently their predictions are not independent: c.f. e.g. Noirhomme et al., 2014 (NeuroImage: Clinical).This arguably minor concern is reinforced by the fact that single-trial-based statistics can dramatically increase the degrees of freedom, and can thus lead to unreasonably confident p-values. Second-level stats across subjects only would be a more conservative approach. Again, the authors did not follow that recommended route.That being said, these statistical issues should not undermine the methodological and neuroscientific work. While I am relatively unhappy with the current answer, the paper remains interesting and potentially important.Consequently, I will not oppose its acceptation for publication, but would strongly recommend the authors to provide a clear code of their analyses and simulations to facilitate future replications.

We agree with the reviewer that the statistical analysis that we have used to assess the main imagery effect is not ideal. As outlined in the previous revision, we still believe this is the most appropriate test for our data because it allows for variation in the intercept between trials which can capture the large variation in onset of imagery between trials. However, to take away the worry that our effect is only significant because of the large degrees of freedom or the violation of the independence assumption, we have decided to run additional between-subject LMM models and have added the following to the Results sections:

“Finally, because we used cross-validation within subjects to calculate reactivation timing, there is a dependence between trials of the same subject, violating the independence assumption of first-order statistical tests. As a sanity check, we performed a second test that did not require splitting trials: we used LMM models with reactivation times averaged over trials within subjects, and only “subject” as a random variable. In this case, the model containing both a random effect of intercept as well as slope per subject best explained the data (see Supplementary File 1C). This between-subject model still showed a significant main effect of perception time (t(24) = -3.24, p = 0.003) with a negative slope (B0 = 2.05, SD = 0.03, B1 = -1.19, SD = 0.37) confirming that the effect was not dependent on between-trial statistics.”

Reviewer #3:I would like to thank the authors for the convincing and thorough revision.I just have two remaining comments.– Related to your answer "We acknowledge that this breaks the independence assumption but we still believe that this is the most valid test for our data, since testing across subjects ignores the large between-trial variability." Can you add a sentence somewhere in the text about this?

Please see the response to reviewer 2 above.

Just as a comment here, using non-parametric methods like in the EEGLAB LIMO toolboxwould address this issue: e.g. Pernet et al., 2011 (Computational Intelligence and Neuroscience).– The proper URL for MNE software is https://mne.tools/ and the relatedpublication is:A. Gramfort, M. Luessi, E. Larson, D. Engemann, D. Strohmeier, C. Brodbeck, L. Parkkonen, M. Hämäläinen, MNE software for processing MEG and EEG data, NeuroImage, Volume 86, 1 February 2014, Pages 446-460, ISSN 1053-8119

Thank you, we have added this to the manuscript.